# Iguratimod Ameliorates the Severity of Secondary Progressive Multiple Sclerosis in Model Mice by Directly Inhibiting IL-6 Production and Th17 Cell Migration via Mitigation of Glial Inflammation

**DOI:** 10.3390/biology12091217

**Published:** 2023-09-07

**Authors:** Satoshi Nagata, Ryo Yamasaki, Ezgi Ozdemir Takase, Kotaro Iida, Mitsuru Watanabe, Katsuhisa Masaki, Marion Heleen Cathérine Wijering, Hiroo Yamaguchi, Jun-ichi Kira, Noriko Isobe

**Affiliations:** 1Department of Neurology, Neurological Institute, Graduate School of Medical Sciences, Kyushu University, Fukuoka 812-8582, Japan; 2Section Molecular Neurobiology, Department of Biomedical Sciences of Cells & Systems, University of Groningen, University Medical Center Groningen (UMCG), MS Center Noord Nederland, 9713 AV Groningen, The Netherlands; 3School of Physical Therapy, Faculty of Rehabilitation, Reiwa Health Sciences University, Fukuoka 811-0213, Japan; 4Translational Neuroscience Center, Graduate School of Medicine, and School of Pharmacy at Fukuoka, International University of Health and Welfare, Fukuoka 831-8501, Japan; 5Department of Neurology, Brain and Nerve Center, Fukuoka Central Hospital, Fukuoka 810-0022, Japan

**Keywords:** iguratimod, secondary progressive multiple sclerosis, connexin 47, experimental autoimmune encephalomyelitis, IL-6, astrocyte

## Abstract

**Simple Summary:**

We previously developed a novel model of progressive multiple sclerosis, called progressive experimental autoimmune encephalomyelitis, in mice with oligodendroglia-specific knockout of the connexin-47 gene. Our previous research showed that iguratimod, an antirheumatic drug, effectively ameliorated acute experimental autoimmune encephalomyelitis. In this study, iguratimod was administered to mice with progressive experimental autoimmune encephalomyelitis, which resulted in improvement of clinical severity, reduced demyelination, and decreased glial inflammation. Interleukin-6 levels and T helper 17 cell infiltration were also reduced. Furthermore, T helper 17 cell migration and interleukin-6 production in cell culture were inhibited by iguratimod. In conclusion, iguratimod successfully mitigated clinical signs of progressive experimental autoimmune encephalomyelitis by suppressing T helper 17 migration and inhibiting interleukin-6 production in proinflammatory-activated glial cells.

**Abstract:**

We previously reported a novel secondary progressive multiple sclerosis (SPMS) model, progressive experimental autoimmune encephalomyelitis (pEAE), in oligodendroglia-specific *Cx47*-inducible conditional knockout (*Cx47* icKO) mice. Based on our prior study showing the efficacy of iguratimod (IGU), an antirheumatic drug, for acute EAE treatment, we aimed to elucidate the effect of IGU on the SPMS animal model. We induced pEAE by immunizing *Cx47* icKO mice with myelin oligodendrocyte glycoprotein peptide 35–55. IGU was orally administered from 17 to 50 days post-immunization. We also prepared a primary mixed glial cell culture and measured cytokine levels in the culture supernatant after stimulation with designated cytokines (IL-1α, C1q, TNF-α) and lipopolysaccharide. A migration assay was performed to evaluate the effect of IGU on the migration ability of T cells toward mixed glial cell cultures. IGU treatment ameliorated the clinical signs of pEAE, decreased the demyelinated area, and attenuated glial inflammation on immunohistochemical analysis. Additionally, IGU decreased the intrathecal IL-6 level and infiltrating Th17 cells. The migration assay revealed reduced Th17 cell migration and IL-6 levels in the culture supernatant after IGU treatment. Collectively, IGU successfully mitigated the clinical signs of pEAE by suppressing Th17 migration through inhibition of IL-6 production by proinflammatory-activated glial cells.

## 1. Introduction

Multiple sclerosis (MS) is an inflammatory demyelinating disease of the central nervous system (CNS), and its pathological targets are thought to be myelin proteins or oligodendrocytes. More than 80% of MS patients demonstrate a relapsing remitting clinical course, but some proportion of patients undergo a secondary progressive phase, and worsening of neurologic function is observed independent from relapses [1,2]. Connexin (Cx) is a component of gap junctions, which connect the cytoplasm between cells. In the CNS, oligodendroglia express Cx29, Cx32, and Cx47 [3,4,5,6,7], while astroglia express Cx26, Cx30, and Cx43 [8,9,10,11,12]. We previously reported loss of oligodendroglial Cx32 and Cx47 expression in most active and chronic lesions from all analyzed cases of MS and neuromyelitis optica spectrum disorder [13]. Recently, we reported a novel model of secondary progressive multiple sclerosis (SPMS) induced by immunization with myelin oligodendrocyte glycoprotein peptide 35–55 (MOG_35–55_) in oligodendroglia-specific *Cx47*-inducible conditional knockout (*Cx47* icKO) mice [14]. This new progressive experimental autoimmune encephalomyelitis (pEAE) model demonstrated clinical exacerbation during the acute phase, followed by a progressive course with more pronounced demyelination than that in *Cx47* floxed/floxed (*fl*/*fl*) mice and is considered to be a suitable model for SPMS. In pEAE mice, infiltration of T helper 17 (Th17) cells into lumbar lesions was enhanced, and proinflammatory “A1” astroglia and injury-related microglia were increased in the spinal cord during the chronic phase [14,15]. 

Iguratimod (IGU), a selective cyclooxygenase-2 (COX-2) inhibitor [16], is in clinical use for treatment of rheumatoid arthritis (RA) with low risk of progressive multifocal leukoencephalopathy and other adverse effects. Recently, several studies have shown that IGU exerts various effects on the immune system. For instance, IGU can inhibit interleukin (IL)-6 or IL-8 in RA synovial fibroblast-like cells via interference with nuclear factor (NF)-κB [17,18], IL-17-mediated signaling by disrupting Act1 or migration inhibitory factor (MIF) [19], and regulation of B cell differentiation by disrupting the protein kinase C (PKC) pathway [20]. We previously reported that IGU ameliorates acute and chronic EAE by suppressing microglia and macrophages through inhibition of the NF-κB pathway [21]. Astroglia also activate microglia [15], but the direct effect of IGU on astroglia is unclear. Our aim is to clarify whether IGU ameliorates the clinical course of pEAE and elucidate the biological mechanisms of IGU.

## 2. Materials and Methods

### 2.1. Ethics Statement

All efforts were made to minimize the number and suffering of mice based on the guidelines for the proper conduct of animal experiments published by the Science Council of Japan and the Animal Research: Reporting of In Vivo Experiments (ARRIVE) guidelines 2.0 for animal research [22,23]. The Animal Care and Use Committee of Kyushu University granted ethical approval for the study on 28 January 2021 (#A21–183) and permitted an extension of the experimental period until 31 March 2023 (#A23–059).

### 2.2. Generation of PLP/CreER^T^;Cx47^fl/fl^ Mice

To generate *Cx47* conditional floxed mice (*Cx47^fl^*^/*fl*^ mice), targeting vectors for the “knockout-first allele” [24] were obtained from the European Conditional Mouse Mutagenesis Program (ID: 45393). The targeting vector consisted of an *En2SA-IRES-LacZ* reporter cassette, a neomycin resistance cassette, and *loxP* sites flanking an exon of the *Cx47* gene (NCBI Gene ID: 118454). The targeting vector was linearized and introduced into RENKA embryonic stem (ES) cells (C57BL/6) by electroporation. After selection with geneticin (G418 sulfate), the resistant clones were isolated, and their deoxyribonucleic acid (DNA) was screened for homologous recombination by polymerase chain reaction (PCR) assays using the following primer set: *sc_3GR3n*, 5′-CTT ATA GGC TGG GAC TTG TGG ATG GC-3′ and *neo_G02*, 5′-ATC AGG ACA TAG CGT TGG CTA C-3′. The PCR-positive ES clones were expanded, and their isolated DNA was further analyzed by PCR amplification using the following primer sets: *sc_5GF1*, 5′-GCT TCA TGG ATC AGG GTA ATT CCA G-3′ and *LAR3*, 5′-CAC AAC GGG TTC TGT TAG TCC-3′ for 5′ amplification; *sc_3GR4n*, 5′-TCC TGA GGT AAC CCT AAC AAA CAC G-3′ and *neo_MS02*, 5′-TTC GCA GCG CAT CGC CTT CTA TCG-3′ for 3′ amplification; and *loxF3*, 5′-GAG GCT GAG ATG GCG CAA CG-3′ and *loxR2*, 5′-TTC AGT CAT CCG CGC TAC ACA CC-3′ for amplification of the 3′ *loxP* region. Homologous recombination of the clones was confirmed by genomic Southern hybridization with probing for the neomycin resistance gene. Homologous recombinant ES cell clones were aggregated with eight-cell ICR mouse embryos to generate chimeric mice. To obtain the floxed allele by removing the reporter gene and neomycin resistance cassette, the chimeric mice were crossed with *CAG-Flp* transgenic mice (CARD, Kumamoto, Japan) that expressed Flp in their germ cells [25]. The floxed allele was identified by PCR using the following primer set: *F14989*, 5′-TGG TGC TGG AAT TGG AAG C-3′ and *R15381*, 5′-TCT GCA TTC ACA TCC TAT GTG C-3′. *Cx47^fl^*^/*fl*^ mice and *PLP*/*CreER^T^* mice (Jackson Laboratory, Bar Harbor, ME, USA), which is a useful mouse line for inducible ablation of target genes in mature oligodendroglia [26], were bred to generate *PLP*/*CreER^T^*;*Cx47^fl^*^/*fl*^ mice. The following primer sets were used for mouse genotyping: *mGjc2-F3*, 5′-TGG GCT CAA TGC AAC CTC TC-3′ and *mGjc2-R1*, 5′-CAG GGT TTG GTC TCC AGC TT-3′ were used for WT mice and yielded a 1172 bp band; *loxF3*, 5′-GAG GCT GAG ATG GCG CAA CG-3′ and *loxR2*, 5′-TTC AGT CAT CCG CGC TAC ACA CC-3′ were used for *Cx47^fl^*^/*fl*^ mice and yielded a 202 bp band; *olMR1084*, 5′-GCG GTC TGG CAG TAA AAA CTA TC-3′ and *olMR1085*, 5′-GTG AAA CAG CAT TGC TGT CAC TT-3′ were used for *PLP*/*CreER^T^* mice and yielded a 100 bp band.

### 2.3. Tamoxifen Injection 

At 8–10 weeks of age, *PLP*/*CreER^T^*;*Cx47^fl^*^/*fl*^ mice were intraperitoneally injected with 1 mg of tamoxifen (Sigma-Aldrich, Steinheim, Germany), which drives the *Cre*/*loxP* estrogen receptor transgenic system [27], in 100 µL of corn oil twice per day for 5 days. 

### 2.4. Induction and Clinical Evaluation of EAE 

EAE was induced by subcutaneous injection of MOG_35–55_ peptide (4 mg/mL, S-PEP; Scrum, Tokyo, Japan) emulsified in complete Freund’s adjuvant containing 10 mg/mL *Mycobacterium tuberculosis* H37RA (#7027; Chondrex Inc., Woodinville, WA, USA) at a dose of 200 µg (100 µL) per mouse, followed by intraperitoneal injection with pertussis toxin (300 ng per mouse, #168-22471; Wako, Osaka, Japan) on days 0 and 2. Mice were examined daily for signs of EAE and scored as follows: 0, no disease; 1, limp tail; 2, abnormal gait and hind limb weakness (shaking); 3, paralysis of the two hind limbs; 4, tetraplegia; 5, moribund (death). 

### 2.5. IGU Treatment In Vivo 

IGU (Toyama Chemical Co. Ltd., Tokyo, Japan) was suspended in 100 µL of 0.5% (*w*/*v*) methylcellulose solution and orally administered twice daily (50 mg/kg/day) from 17 to 50 dpi for therapeutic treatment of mice with EAE. The same amount of methylcellulose solution was administered to vehicle-treated mice.

### 2.6. Tissue Preparations 

The mice underwent deep anesthesia using isoflurane (Pfizer Japan Inc., Tokyo, Japan) and were then subjected to transcardial perfusion with phosphate-buffered saline (PBS) followed by 4% paraformaldehyde (PFA) in 0.1 M PBS. The spinal cords were carefully separated and fixed overnight in 4% PFA at 4 °C. Subsequently, the tissues were processed into paraffin sections of 5 µm thickness. For preparing frozen sections, the spinal cords were collected and fixed overnight in 4% PFA following the same procedure. They were then sequentially incubated with 20% and 30% sucrose in PBS solution for 24 h each at 4 °C. The resulting tissues were embedded in Tissue-Tek optimal cutting temperature compound (4583, SAKURA, Torrance, CA, USA) and stored at −80° C.

### 2.7. Histopathological and Immunohistochemical Analyses 

A portion of the paraffin-embedded axial L4–5 spinal cord sections were examined by immunostaining using an indirect immunoperoxidase method. After deparaffinization, the endogenous peroxidase was neutralized by treating the sections with 0.3% hydrogen peroxide in absolute methanol for 30 min. Following that, the sections underwent washing with Tris-HCl for 5 min, followed by immersion in 10 mM citrate buffer and autoclaving at 120 °C for 10 min. All sections were then cooled to room temperature (RT) and incubated overnight at 4 °C with primary antibodies. The primary antibody dilution solution contained 5% normal goat serum and 1% bovine serum albumin in 50 mM Tris-HCl (pH 7.6). The next day, the slides were rinsed, and the sections were labeled either with a streptavidin-biotin complex or using an enhanced indirect immunoperoxidase method with Envision (K4003, Dako, Glostrup, Denmark). For color reactions, 3,3-diaminobenzidine tetrahydrochloride (D5637, Sigma-Aldrich, Tokyo, Japan) was employed, and the sections were finally counterstained with hematoxylin. Other paraffin sections were kept for immunofluorescence analysis. For the immunofluorescence analysis, the lumbar spinal cord sections were deparaffinized, washed, and autoclaved. The sections were then incubated with primary antibodies overnight at 4 °C. Next, the sections were washed and incubated with secondary antibodies conjugated with Alexa Fluor 488 or 594 (1:1000; Thermo Fisher, Rockford, IL, USA) and 4,6-diamidino-2-phenylindole (DAPI) (Sigma-Aldrich, Tokyo, Japan). Finally, the sections were mounted with PermaFluor (#TA-030-FM; Thermo Scientific, Waltham, MA, USA). For frozen spinal cord sections, sectioning was performed using a Leica CM 1850 cryostat (Leica Microsystems GmbH, Wetzlar, Germany), and after incubation in a blocking solution (PBS-T with 10% normal goat serum) for 2 h at RT, the sections were incubated at 4 °C with primary antibodies overnight. Subsequently, the sections were incubated with secondary antibodies conjugated with Alexa Fluor 488 or 594 (1:1000; Thermo Fisher, Rockford, IL, USA) and DAPI (Sigma-Aldrich, Tokyo, Japan), and mounted with PermaFluor (#TA-030-FM; Thermo Scientific, Waltham, MA, USA). Images were captured using a confocal laser microscope system (Nikon A1; Nikon, Tokyo, Japan). Additional information regarding the primary antibodies used is provided in Appendix A.

### 2.8. Quantification of Immunohistochemical Images 

The immunohistochemically stained sections were subjected to automated scanning and quantification using ImageJ Analysis 1.51h software “https://imagej.nih.gov/ij/index.html (accessed on 5 September 2018)”. For quantifying immunostaining, with the exception of Iba-1 staining, transverse spinal cord sections were divided by horizontal and vertical lines passing through the central canal (solid lines in Appendix A). The central gray matter area was manually excluded, defining the white matter region of interest (the area encircled by a dashed line in Appendix A). The positively stained areas within this region were automatically measured and expressed as a percentage of the white matter area of interest. In the case of quantifying Iba-1 staining, transverse spinal cord sections were divided by horizontal and vertical lines passing through the central canal (solid lines in Appendix A). The region of interest encompassing the central gray matter and white matter (the area encircled by a dashed line in Appendix A) was defined. The positively stained areas within this region were measured. Additionally, quantification was performed on L4–5 spinal cord sections that were immunostained with each antibody. The stained areas were expressed as a percentage of the total region of interest. The results for each experimental condition were averaged from four to six sections per mouse. 

### 2.9. Microglial Circularity Analysis 

The circularity of microglial cells (calculated as circularity = 4πS/L^2^) was automatically determined using ImageJ software. Cells exhibiting circularity values close to 1 were classified as having a round morphology, indicating an activated state [28].

### 2.10. Flow Cytometry 

Spinal cord and brain cells were isolated using a density gradient technique following a previously described method [29]. In brief, the spinal cord and brain tissues were minced with a tissue homogenizer to obtain a single-cell suspension. The cell suspension was mixed with stock isotonic Percoll (diluted 10-fold in 10× Hanks’ balanced salt solution (HBSS) without Ca^2+^ and Mg^2+^) to create a 30% Percoll cell solution. A 70% Percoll gradient was then carefully layered beneath the 30% Percoll cell solution, and the mixture was subjected to centrifugation at 800× *g* for 40 min. Subsequently, the myelin layer was discarded, and the mononuclear cell interphase was isolated and resuspended in FACS buffer. To measure the three principal CD4^+^ T cell subsets (Th1, Th17, and Th1/17), monoclonal antibodies against surface chemokine receptors were used, following a previously described protocol [30]. For surface marker staining, the cells were incubated with fluorochrome-conjugated antibodies targeting CD4, CXCR3, and CCR6 for 30 min at 4 °C. Subsequently, the cells were analyzed using a FACS Verse™ Flow Cytometer (Becton Dickinson, Tokyo, Japan). The percentages of CXCR3^+^ CCR6^−^ CD4^+^ T cells corresponding to Th1 cells, CXCR3^−^ CCR6^+^ CD4^+^ T cells corresponding to Th17 cells, and CXCR3^+^ CCR6^+^ CD4^+^ T cells corresponding to Th1/Th17 cells in CD4^+^ T cells were measured. 

### 2.11. Isolation of Splenocytes 

Spleens were extracted aseptically from mice with acute-phase EAE at 17 dpi and dissociated into individual cells following a previously described method [14]. In brief, splenocytes were isolated from EAE mice and passed through a 100 µm cell strainer to obtain a single-cell suspension. The cells were then subjected to centrifugation at 300× *g* for 5 min. After discarding the supernatant, PBS was added, and the cells underwent centrifugation as mentioned above. Subsequently, the cells were resuspended in red blood cell lysis buffer, incubated for 10 min, and then centrifuged for 5 min at 300× *g*. Viable cells were counted using a hemocytometer and 0.4% trypan blue staining.

### 2.12. T Cell Proliferation Assay 

Mouse spleens were collected near the peak of acute EAE. Splenocyte suspensions (7.5 × 10^5^ cells/100 µL/well) were placed in a 96-well plate, and 100 µL of Roswell Park Memorial Institute 1640 medium containing 0 (negative control), 2.5, 12.5, or 25 µg/mL MOG_35–55_ was added. The plates were incubated at 37 °C in a humidified atmosphere containing 5% CO_2_ for 72 h. T cell proliferation was assessed using a bromodeoxyuridine (BrdU) kit (ab126556; BrdU cell proliferation enzyme-linked immunosorbent assay (ELISA) kit, Abcam, Cambridge, UK) following the manufacturer’s instructions. In brief, during the last 24 h of the 72 h incubation period, 20 µL of 1× BrdU solution was added to each well. A background control without BrdU was included in the plate. After 72 h of incubation, the cells were fixed with 200 µL of fixing solution for 30 min and then incubated with 100 µL of anti-BrdU monoclonal detector antibody for 1 h at RT. The cells were washed and incubated with 100 µL of 1× peroxidase-conjugated goat anti-mouse IgG, which had been filtered through a 0.22-µm syringe filter, for 30 min at RT. The cells were then incubated with 100 µL of tetramethylbenzidine peroxidase in the dark for 30 min. After adding 100 µL of the stop reaction solution, the absorbances of the solutions in the wells were measured using a microplate reader (MTP-800AFC; Corona Electric, Hitachinaka, Japan).

### 2.13. Glial Cell Cultures 

Primary mixed glial cell cultures were prepared from the brains of newborn C57BL/6 J mice using a previously described method [31,32]. Briefly, under sterile conditions, brains were removed and the meninges were carefully removed using tweezers. The tissue was dissociated by passing it through a nylon mesh in HBSS (Sigma-Aldrich, Saint Louis, MO, USA) containing 50 U/mL penicillin and 50 µg/mL streptomycin (Gibco, Thermo Fisher Scientific, Waltham, MA, USA) to prevent contamination. After being washed with HBSS, the cell suspension was plated in 75 cm^2^ culture flasks at a density of one to two brains per flask in 10 mL of culture medium. The culture medium consisted of Dulbecco’s Modified Eagle’s Medium (Sigma-Aldrich, Tokyo, Japan) supplemented with 10% fetal bovine serum (Equitech-Bio, Kerrville, TX, USA), 5 µg/mL bovine insulin (Sigma-Aldrich, Tokyo, Japan), and 0.2% glucose. The cells were maintained at 37 °C in a humidified atmosphere containing 5% CO_2_. During the first week, the medium was changed three times, while during the second week, no medium change was performed to stimulate the proliferation of microglia. Upon reaching confluency (12–15 days), mixed glial cells were detached using Accutase (Innovative Cell Technologies, San Diego, CA, USA) treatment and then replated. After an additional 5–10 days in culture, mixed glial cell cultures, which had reached 100% confluence, were ready for experiments.

### 2.14. Harvesting of Glial Cell Culture Supernatant 

To measure cytokine or chemokine production, glial cells were incubated with stimulants including 1 µg/mL LPS, 3 ng/mL IL-1α, 30 ng/mL TNF-α, and 400 ng/mL C1q at 37 °C in a humidified atmosphere containing 5% CO_2_ for 24 h. Glial cells were simultaneously treated with or without IGU. During the incubation, 50 µL of each supernatant was harvested at 2, 4, 6, 8, and 24 h, centrifuged, and stored at −80 °C.

### 2.15. Multiplexed Fluorescence Immunoassay for Cytokines 

Cytokine levels in the supernatant were assessed using a Bio-plex Pro™ Assay (M60-009RDPD; Bio-Rad, Tokyo, Japan) following the manufacturer’s instructions. The concentrations of various cytokines and chemokines (IL-1α, IL-1β, IL-4, IL-6, IL-10, IL-17, IFN-γ, CCL2/monocyte chemotactic protein (MCP)-1, CCL3/MIP-1α, CCL5/regulated on activation, normal T cell expressed and secreted (RANTES)) in the supernatants were quantified using a multiplexed fluorescence bead-based immunoassay as described previously [32]. To calculate the cytokine and chemokine levels, a standard curve was generated using standards that underwent the same procedure as the supernatant samples. The limit of detection for each molecule was determined based on the recovery of the corresponding standard, and the lowest value with >70% recovery was defined as the lower limit of detection. Some samples exceeded the upper limit of detection, and other samples fell below the lower limit of detection.

### 2.16. Chemokine Analysis by ELISA 

Chemokine (CCL-20/MIP-3α) levels in mixed glial cell culture supernatants were measured using a Mouse CCL-20/MIP-3 alpha Quantikine ELISA Kit (MCC200, R&D Systems, Minneapolis, MN, USA) according to the manufacturer’s instructions. Briefly, 100 µL of the samples was added to each well and incubated for 2 h. After washing, 200 µL of human MIP-3α conjugate was added to each well and incubated for 2 h. The wells were washed, and 200 µL of substrate solution was added to each well and incubated. After addition of 100 µL of the stop reaction solution, the absorbances of the solutions in the wells were measured using a microplate reader. 

### 2.17. Migration Assay 

Transwell chambers (Corning, Corning, NY, USA) consisted of an upper chamber and a lower chamber with or without mixed glial cells (1.0 × 10^6^ cells/well), stimulants (IL-1α (3 ng/mL), C1q (400 ng/mL), TNF-α (30 ng/mL), LPS (1 µg/mL)), and IGU (10 µg/mL). CD4^+^ T cells (5.0 × 10^5^ cells/well) from acute EAE mice were loaded in the upper chamber. After 24 h of incubation, cells that had migrated were counted using flow cytometry as described above. Three principal CD4^+^ T cell subsets (Th1, Th17, and Th1/17) were measured using monoclonal antibodies against surface chemokine receptors, as previously described [30]. For surface marker staining, cells were incubated with fluorochrome-conjugated antibodies against CD4, CXCR3, and CCR6 for 30 min at 4 °C and analyzed using a FACS Verse™ Flow Cytometer (Becton Dickinson). The absolute counts of CXCR3^+^ CCR6^−^ CD4^+^ T cells corresponding to Th1 cells, CXCR3^+^ CCR6^+^ CD4^+^ T cells corresponding to Th1/Th17 cells, and CXCR3^−^ CCR6^+^ CD4^+^ T cells corresponding to Th17 cells among CD4^+^ T cells were measured. 

### 2.18. Incubation of CD4^+^ T Cells with IGU 

CD4^+^ T cells from acute EAE mice were incubated with or without IGU for 24 h. CD4^+^ T cells were counted by flow cytometry using the same procedure as described above. For surface marker staining, cells were incubated with fluorochrome-conjugated antibodies against CD4, CXCR3, and CCR6 for 30 min at 4 °C and analyzed using a FACS Verse™ Flow Cytometer (Becton Dickinson). The absolute counts of CXCR3^+^ CCR6^−^ CD4^+^ T cells corresponding to Th1 cells, CXCR3^+^ CCR6^+^ CD4^+^ T cells corresponding to Th1/Th17 cells, and CXCR3^−^ CCR6^+^ CD4^+^ T cells corresponding to Th17 cells among CD4^+^ T cells were measured.

### 2.19. Statistical Analysis

All data are presented as the mean ± standard error of the mean. The statistical significance of differences between values was determined using an unpaired *t*-test, one-way analysis of variance, or two-way analysis of variance. Values of *p* < 0.05 were considered statistically significant. 

## 3. Results

### 3.1. IGU Ameliorated the Clinical Severity and Demyelination in the Chronic Phase of pEAE

To investigate the effect of IGU in pEAE mice, we induced EAE in *PLP*/*CreER^T^*;*Cx47^fl^*^/*fl*^ mice. First, we intraperitoneally injected tamoxifen at 8–10 weeks of age (day −14 to −10) to obtain *Cx47* icKO mice in which *Cx47* was ablated only in oligodendroglia. Then, *Cx47* icKO mice were immunized with MOG_35–55_ to induce pEAE on day 0. IGU (50 mg/kg, twice a day) or methylcellulose (vehicle) was orally administered from 17 to 50 days post-immunization (dpi) (Figure 1a). After IGU treatment, clinical signs of pEAE were significantly ameliorated in IGU-treated mice during the chronic phase (*p* < 0.01 at 26–29 dpi, *p* = 0.0084 at 31 dpi, *p* < 0.05 at 34–50 dpi; Figure 1a, red polygonal line) compared with those in vehicle-treated mice (blue polygonal line). The areas under the curve of clinical scores were also significantly decreased by IGU treatment (*p* < 0.0001; Figure 1b). Immunohistochemical examinations performed at 50 dpi revealed that IGU decreased areas of white matter demyelination in the lumbar spinal cord compared with vehicle-treated mice (*p* = 0.0052; Figure 1c,d).

### 3.2. IGU Inhibited Inflammatory Cell Infiltration of the Lumbar Spinal Cord in the Chronic Phase of pEAE

Next, we compared the areas of CD3^+^ T cells, F4/80^+^ macrophages, glial fibrillary acidic protein (GFAP)^+^ astroglia, and Iba-1^+^ microglia in the lumbar spinal cord between IGU-treated and vehicle-treated pEAE mice (Figure 2a–i). Each area was significantly smaller in IGU-treated mice than in vehicle-treated mice (*p* = 0.0076 for CD3^+^ T cells, Figure 2a,b; *p* = 0.0256 for F4/80^+^ macrophages, Figure 2c,d; *p* = 0.0190 for GFAP^+^ astroglia, Figure 2e,f; *p* = 0.0070 for Iba-1^+^ microglia, Figure 2g,h). In vehicle-treated pEAE mice, Iba-1^+^ microglia had enlarged cell bodies and a few short processes, indicating that the microglia were in an activated (reactive) state (amoeboid or bushy phenotype). Meanwhile, in IGU-treated pEAE mice, Iba-1^+^ microglia had thin soma and radially projecting thin processes, indicating that the microglia were in the resting (homeostatic) state (ramified phenotype) [33] (*p* = 0.0173; Figure 2i). These results indicated that IGU suppresses immunocyte infiltration and mitigates glial inflammation in the lesion.

### 3.3. IGU Reduced A1 Astroglia and M1-Like Microglia during the Chronic Phase

Next, we performed immunohistochemical examinations of the astroglial and microglial phenotype during the chronic phase (50 dpi). The areas with proinflammatory C3^+^ GFAP^+^ A1 astroglia and inducible nitric oxide synthase (iNOS)^+^ Iba-1^+^ M1-like microglia in the lumbar spinal cord were smaller in IGU-treated pEAE mice than in vehicle-treated pEAE mice (*p* = 0.0276 for C3^+^ GFAP^+^ A1 astroglia, Figure 3a,b; *p* = 0.0140 for iNOS^+^ Iba-1^+^ M1-like microglia, Figure 3c,d). However, the areas of anti-inflammatory S100A10^+^ GFAP^+^ A2 astroglia and Arg-1^+^ Iba-1^+^ M2-like microglia in the lumbar spinal cord were comparable between IGU-treated and vehicle-treated pEAE mice (*p* = 0.7883 for S100A10^+^ GFAP^+^ A2 astroglia, Figure 3e,f; *p* = 0.5045 for Arg-1^+^ Iba-1^+^ M2-like microglia, Figure 3g,h). Thus, IGU reduced proinflammatory A1 astroglial and M1-like microglial inflammation in the lumbar spinal cord during the chronic phase. 

### 3.4. IGU Suppressed Th17 Cell Migration in the Spinal Cord and Cerebrospinal Fluid (CSF) IL-6 Production during the Chronic Phase

Next, we performed immunohistochemical examinations, flow cytometric analysis, and CSF cytokine analysis in pEAE mice. The IL-17A^+^ cell area in the lumbar spinal cord was smaller in IGU-treated pEAE mice than in vehicle-treated pEAE mice at 50 dpi (*p* = 0.0037; Figure 4a,b). The number of IL-17A^+^ CD3^+^ Th17 cells was also decreased in IGU-treated pEAE mice (*p* = 0.0140; Figure 4c,d). Furthermore, we performed flow cytometric analysis using CNS tissue (spinal cord and brain) of pEAE mice in the chronic phase. We identified the helper T cell lineage based on chemokine receptor expression. The flow cytometric analysis revealed that the proportion of C-X-C motif chemokine receptor (CXCR)3^−^ CCR6^+^ Th17 cells among CD4^+^ cells was significantly decreased in the IGU-treated group (*p* = 0.0286; Figure 5a–c). However, the proportions of CXCR3^+^ CCR6^+^ Th1/Th17 cells and CXCR3^+^ CCR6^−^ Th1 cells among CD4^+^ cells were comparable between IGU-treated and vehicle-treated mice (*p* = 0.3429 for CXCR3^+^ CCR6^+^ CD4^+^ Th1/Th17 cells, Figure 5d; *p* = 0.3429 for CXCR3^+^ CCR6^−^ CD4^+^ Th1 cells, Figure 5e). A cytokine analysis of the mouse CSF showed that the IL-6 level was lower in IGU-treated mice than in vehicle-treated mice at 50 dpi (*p* = 0.0023; Figure 5f), while the IL-1β, IL-4, and interferon (IFN)-γ levels were comparable between the two groups (Figure 5g–i). 

### 3.5. IGU Inhibited IL-6 and CCL2 Release from Glial Cells in the In Vitro Activated Glial Inflammation Model 

Next, we prepared in vitro mixed glial cell culture containing astroglia and microglia from newborn wild-type (WT) mice. The culture was altered to produce proinflammatory A1 astroglia and M1-like microglia by the addition of stimulating cytokines (IL-1α, C1q, tumor necrosis factor (TNF)-α, and lipopolysaccharide (LPS)) [15] to simulate the glial cell conditions in pEAE mice. We measured the cytokine levels in the culture supernatant at 2, 4, 6, 8, and 24 h after administration of stimulating cytokines with or without IGU treatment (Figure 6). IL-1β levels at 24 h, IL-6 levels at 6–24 h, and C-C motif chemokine ligand (CCL)2 levels at 24 h post-stimulation in the culture supernatant were decreased in the IGU-treated group (*p* = 0.0460 for IL-1β levels at 24 h, Figure 6b; *p* = 0.0066 for IL-6 levels at 6 h, *p* < 0.0001 for IL-6 levels at 8 h, *p* = 0.0016 for IL-6 levels at 24 h, Figure 6d; *p* = 0.0058 for CCL2 levels at 24 h, Figure 6g). The CCL20 levels in the culture supernatant at 24 h post-stimulation were comparable between the IGU- and vehicle-treated groups. Collectively, IL-6 levels were decreased by IGU treatment both in vivo and in vitro. 

### 3.6. IGU Decreased the Number of Th17 Cells Migrating toward the Glial Cell Culture in the In Vitro Migration Assay

The composition of the glial cell culture supernatant is essential for the migration and infiltration of pathogenic T cells into the lesion. Next, to clarify whether IGU alters the supernatant composition, we performed an assay to assess CD4^+^ T cell migration toward the proinflammatory mixed glial cell culture (Figure 7). We induced proinflammatory mixed glial cells (Mix) by administering cytokines (IL-1α, C1q, TNF-α, and LPS) (Stim) with or without IGU in the lower chamber, and CD4^+^ T cells isolated from conventional EAE mice were placed in the upper chamber (Appendix A). After the chamber was incubated for 24 h, we assessed the migration of three principal pathogenic CD4^+^ T cell subsets (Th17, Th1/Th17, and Th1) from the upper chamber to the lower chamber using flow cytometry (Figure 7b). The number of migrating CXCR3^−^ CCR6^+^ CD4^+^ Th17 cells, which was increased by cytokine stimulation, was decreased by IGU treatment (*p* < 0.001 for the Mix(+) Stim(+) IGU(−) vs. Mix(+) Stim(+) IGU(+) conditions, Figure 7c). Although cytokine stimulation increased the proportion of CXCR3^+^ CCR6^+^ CD4^+^ Th1/Th17 cells, IGU did not alter the migration of these cells (*p* = 0.7394 for the Mix(+) Stim(+) IGU(−) vs. Mix(+) Stim(+) IGU(+) conditions, Figure 7d). Neither cytokine stimulation nor IGU affected the cell number and migration status of CXCR3^+^ CCR6^−^ CD4^+^ Th1 cells (Figure 7e). 

### 3.7. IGU Did Not Affect the Differentiation of CD4^+^ T Cells or T Cell Proliferation 

Next, we investigated whether IGU directly affects CD4^+^ T cells. In an in vitro assay, we incubated CD4^+^ T cells magnetically separated from the spleens of conventional EAE mice with or without IGU treatment for 24 h, and no significant difference in the proportion of Th17, Th1/Th17, or Th1 cells was found between the groups (Appendix A). Moreover, we performed an in vivo proliferation assay using splenocytes from IGU-treated pEAE mice. Again, there was no difference in the peripheral T cell response caused by IGU treatment when cells were incubated with MOG at different concentrations (0, 2.5, 12.5, 25 µg/mL) (Appendix A). 

## 4. Discussion

We used IGU to treat pEAE mice, which is a novel mouse model of SPMS that presents an aggravated disease course with pronounced glial inflammation, and examined its effects and mechanisms in vivo and in vitro. The main findings in these experiments were as follows. First, IGU was sufficiently effective in mitigating the clinical signs of pEAE and producing a diminished inflammatory astroglial and microglial phenotype of lesions during the chronic phase of pEAE. Second, IGU suppressed Th17 infiltration in the pEAE mouse spinal cord, and we showed that IGU inhibited Th17 migration in vitro by performing a migration assay using CD4^+^ T cells and proinflammatory mixed glial cells under conditions similar to those of glial cells in pEAE mice. Third, IGU reduced CSF IL-6 levels in pEAE mice. Moreover, we revealed a reduction of the IL-6 level in the mixed glial cell culture media after IGU treatment.

The main mechanism of IGU was initially reported to be COX-2 inhibition [16]. IGU has been reported to produce relatively common side effects such as hepatic dysfunction, renal impairment, and gastric ulcers. However, severe adverse effects, such as malignancies, are reported to be rare [34]. Moreover, recent studies have shown that IGU modifies the immune system via several mechanisms. For instance, IGU can inhibit IL-6 and IL-8 in RA synovial fibroblast-like cells by interfering with NF-κB [17,18], IL-17-mediated signaling by disrupting Act1 or MIF [19], and B cell differentiation by disrupting the PKC pathway [20]. Several studies have examined IGU application to EAE models. IGU treatment ameliorated clinical signs of EAE, and in in vitro studies using myelin basic protein-specific T cells, IGU inhibited production of proinflammatory cytokines such as IFN-γ, IL-6, and TNF in T cells [35]. IGU ameliorated acute and chronic EAE by suppressing infiltration and activation of immune cells, especially microglia and macrophages, through inhibition of the NF-κB pathway [21]. 

We previously reported that oligodendroglial *Cx47* icKO mice demonstrated exacerbation of acute and chronic relapsing EAE. In the CNS tissue of *Cx47* icKO EAE mice (pEAE mice in this manuscript), A1 astroglia and M1-like microglia were pathologically increased in both the acute and chronic phases. Moreover, Th17 cells isolated from the CNS tissue were increased in *Cx47* icKO EAE mice [14]. Therefore, we considered IGU to be the most suitable candidate for the treatment of *Cx47* icKO pEAE mice, which demonstrated enhanced infiltration of Th17 cells and activated glial inflammation in lesions. Therapeutic administration of IGU significantly ameliorated the clinical signs of pEAE mice. Immunohistochemical examinations revealed that IGU inhibited A1 astroglial and M1-like microglial activation in pEAE mice during the chronic phase. Immunostaining and flow cytometry also revealed that IGU decreased the infiltration of pathogenic Th17 cells in pEAE mice during the chronic phase. An in vivo mouse CSF cytokine assay revealed that IGU decreased IL-6 levels during the chronic phase of pEAE. IL-6, together with transforming growth factor-β, induces differentiation of Th17 cells from naïve T cells [36,37]. Thus, IL-6 blockade by IGU treatment might impede Th17 differentiation and accumulation in the spinal cord. 

We performed an in vitro primary mixed glial cell culture experiment to determine the direct effects of IGU on glial cells. However, glial cells need to be harvested from newborn mice, and it is difficult to use *Cx47* icKO mice, which require tamoxifen treatment. Therefore, we prepared a mixed glial cell culture containing astroglia and microglia from newborn WT mice. To obtain mixed glial cell culture containing proinflammatory A1 astroglia and M1-like microglia that mimic the conditions in *Cx47* icKO EAE (pEAE) mice, we added stimulating cytokines (IL-1α, C1q, TNF-α, and LPS) to the culture media [15]. Moreover, we added IGU directly to the proinflammatory mixed glial cell culture, which is appropriate because a previous report showed that orally administered IGU levels in the CNS reach approximately 10% of the peripheral blood levels [38]. We found that IGU could reduce Th17 cell migration toward the mixed glial cell culture in the in vitro migration assay. In the natural course of MOG_35–55_-induced conventional EAE, the number of Th17 cells in the CNS peaks earlier than that of Th1 cells [39]. Thus, Th17 cells may play an essential role in autoimmune CNS inflammation, perhaps mainly during the initial phase of the disease [40]. Because, in this study, IGU was administered immediately after the peak of clinical sign (17 dpi), it is unknown whether IGU might affect glial cells in the earlier chronic phase. In fact, the clinical scores of the IGU-treated group improved immediately after starting IGU treatment. Furthermore, in MS patients, Th17 cells increased in the CSF during clinical relapse [41]. Moreover, in peripheral blood from SPMS patients, IL-17-producing peripheral blood mononuclear cells were increased compared with those in healthy controls, and Th17 cells are considered candidate therapeutic targets in SPMS [42]. CCL20 was reported to strongly induce Th17 migration [43]. IGU did not affect the CCL20 level in our mixed glial culture supernatant. However, IGU treatment reduced CCL2 production in mixed glial cells. Although CCL2 is not a Th17 cell-specific chemokine, CCR2, the CCL2 receptor, is expressed in Th17 cells [44]. Therefore, IGU might secondarily suppress Th17 cell migration. 

We also found that IL-6 levels in mixed glial cell culture were significantly decreased by IGU treatment. This result was comparable to the decrease in CSF IL-6 levels produced by IGU treatment. Astroglia are a primary source of IL-6 in the CNS [45]. Meanwhile, previous reports have shown that neurons can produce IL-6 [46], and IGU can reduce IL-6 expression in T cells [35]. Therefore, factors that influence CSF IL-6 levels other than glial cells should be considered. However, in vitro experiments revealed that IGU directly affected glial cells by inhibiting glial cell-induced inflammation and IL-6 release. Although IL-6 could have been produced by both astroglia and microglia in this study, we assumed that IGU treatment affected IL-6 production in astroglia more than that in microglia. This assumption is not only based on a previous study reporting astroglia as a major source of IL-6, but also from our finding that CCL3, which was mainly produced by microglia in the supernatant of mixed glial cell culture, was not affected by IGU treatment [47]. This result was compatible with the decrements of IL-1β and CCL2, both of which were produced by cultured astroglia stimulated by LPS [48,49], in the mixed glial cell culture supernatant after IGU treatment. In this pEAE model, astroglia might have a strong effect on inflammation because Cx47 on oligodendroglia forms oligodendroglia-astroglia gap junction channels with Cx43 on astroglia, and loss of Cx47 increases astroglial Cx43 hemichannels that secrete various bioactive molecules, including proinflammatory cytokines and chemokines [50,51]. Therefore, IGU dramatically ameliorated the severity of EAE by affecting astroglia rather than microglia. 

A possible mechanism of action of IGU on pEAE, based on the results obtained in this study, is summarized in Appendix A. In the acute phase of EAE, pathogenic T cells, including Th17 cells, infiltrate the blood–brain barrier {1}. Then, activated T cells produce chemoattractants and induce a proinflammatory M1-like phenotype in microglia [52,53] {2}. These M1-like microglia induce an A1 phenotype in astroglia [14,15] {3}. In pEAE mice, astroglia and microglia remained activated as A1 and M1-like phenotypes, even during the chronic phase [14] {4}, and IGU can suppress microglial and macrophage activation through the NF-κB pathway [21]. Additionally, we found that IGU can suppress not only proinflammatory M1-like microglia but also A1 astroglia. Moreover, IGU can suppress the production of chemoattractants, especially IL-6 and CCL2, in activated glial cells {5}. Therefore, IGU can decrease the migration of pathogenic T cells, Th17 cells, into the CNS {6}, which ameliorates the clinical signs of pEAE. 

This study has several limitations. First, vascular endothelial cells were not used in the migration assay. Generally, vascular endothelial cells are needed to form typical blood–brain barrier models in migration assays. However, we decided to omit these cells to simplify assays and focus on the direct effects of IGU on glial cells. Moreover, we did not examine the effect of IGU on sheer stress; we used a static model, instead of a flow model, in the migration assay. Additionally, we used a mixed glial cell culture containing astroglia and microglia in this study, but we did not examine each type of glial cell culture. We considered it sufficient to use only mixed glial cell culture to examine the direct effect of IGU on glial cell culture and CD4^+^ T cell migration. However, further in vitro functional assays of single glial cells will complement the present findings. Furthermore, in the flow cytometry analysis using in vivo samples, we did not mention absolute numbers and only reported percentages. Additionally, we identified the helper T cell lineage based not on a lineage-defining cytokine or transcription factor but on chemokine receptor expression. Therefore, further flow cytometry analysis would also enhance the current findings. In addition, the EAE scoring was not conducted by personnel who were blinded to the conditions.

## 5. Conclusions

IGU exerts anti-inflammatory effects by suppressing Th17 cell migration through inhibition of glial inflammation and cytokine/chemokine production in SPMS model mice. IGU is suggested to be a novel therapeutic candidate for SPMS.

## Figures and Tables

**Figure 1 biology-12-01217-f001:**
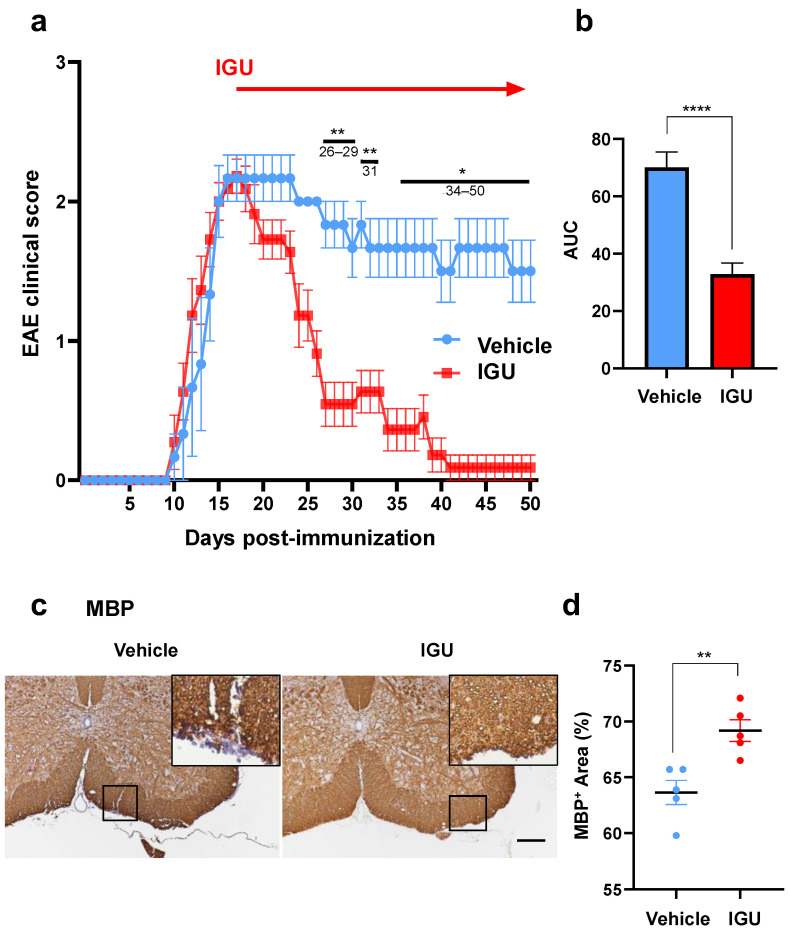
Iguratimod (IGU) ameliorated the severity of progressive experimental autoimmune encephalomyelitis (pEAE) in mice and suppressed demyelination during the chronic phase. (**a**) EAE clinical scores (mean ± standard error of the mean, SEM) of IGU-treated (filled squares, *n* = 10) or vehicle-treated mice (filled circles, *n* = 6). Horizontal bars in the graph indicate the periods with significant differences between the groups. ** 26–29 days post-immunization (dpi), 31 dpi; * 34–50 dpi. (**b**) Comparison of the area under the curve between IGU-treated and vehicle-treated mice. (**c**) Myelin basic protein (MBP) immunostaining in paraffin sections of the lumbar spinal cords of IGU-treated and vehicle-treated mice at 50 dpi. Scale bar: 100 µm. (**d**) The graph shows the percentages of the MBP-positive areas in the spinal cords of IGU-treated and vehicle-treated mice (*n* = 5 per group). Significant differences were determined using an unpaired *t*-test. * *p* < 0.05; ** *p* < 0.01; **** *p* < 0.0001.

**Figure 2 biology-12-01217-f002:**
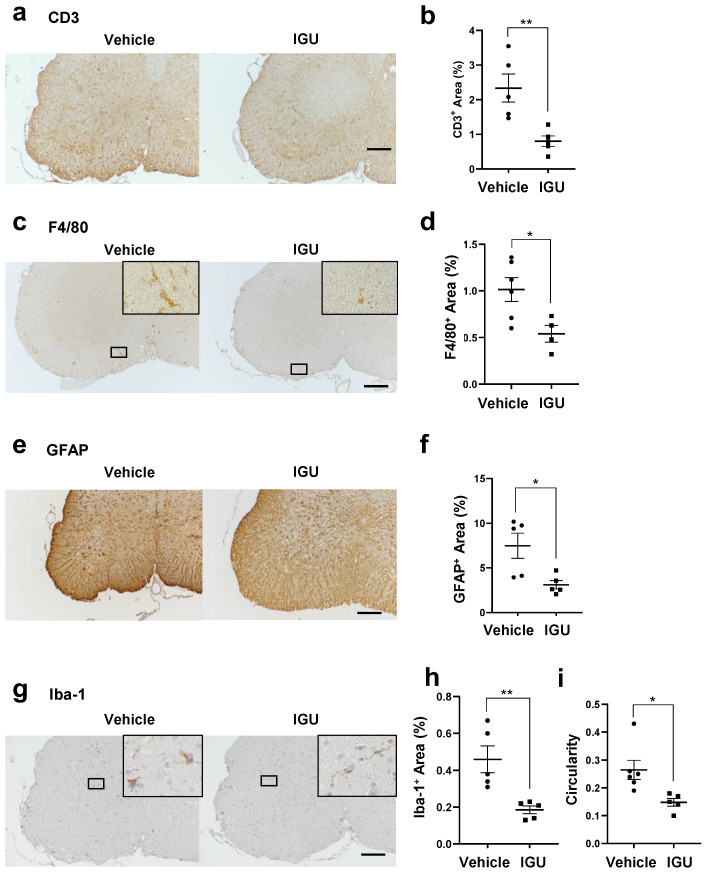
IGU inhibited inflammatory cell infiltration of the lumbar spinal cord during the chronic phase. (**a**,**c**,**e**,**g**) CD3 (**a**), F4/80 (**c**), glial fibrillary acidic protein (GFAP) (**e**), and Iba-1 (**g**) immunostaining in paraffin sections of the lumbar spinal cords of IGU-treated and vehicle-treated mice at 50 dpi. Scale bar: 100 µm. (**b**,**d**,**f**,**h**) The graphs show the percentages of CD3-(*n* = 5 per group) (**b**), F4/80-(*n* = 6, vehicle group; *n* = 4, IGU group) (**d**), GFAP-(*n* = 5 per group) (**f**), and Iba-1-(*n* = 5 per group) (**h**) positive areas in the spinal cords of IGU-treated and vehicle-treated mice. (**i**) Microglial cell circularity quantified at 50 dpi (*n* = 6, vehicle group; *n* = 5, IGU group). Scale bar: 100 µm. Significant differences were determined using an unpaired *t*-test. * *p* < 0.05; ** *p* < 0.01.

**Figure 3 biology-12-01217-f003:**
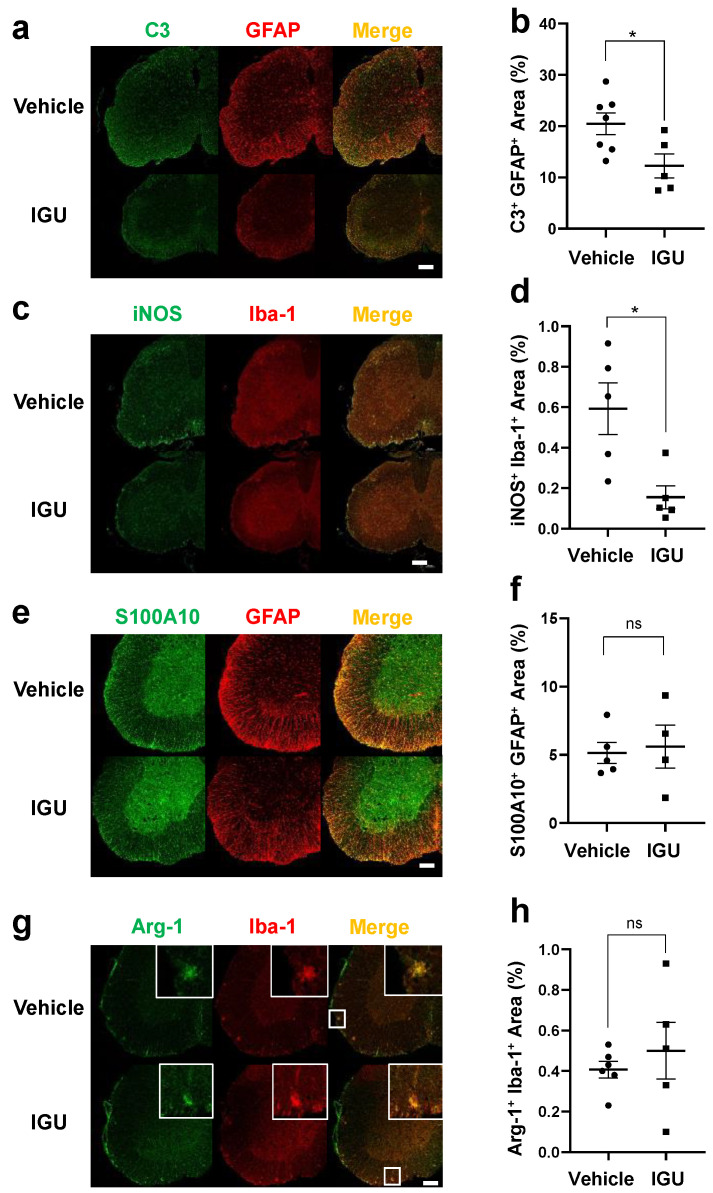
IGU reduced A1 astroglia and M1-like microglia during the chronic phase. (**a**,**c**,**e**,**g**) Double immunofluorescence staining of the lumbar spinal cord for C3 and GFAP (**a**), inducible nitric oxide synthase (iNOS) and Iba-1 (**c**), S100A10 and GFAP (**e**), and Arg-1 and Iba-1 (**g**) at 50 dpi. Scale bar: 100 µm. (**b**,**d**,**f**,**h**) The graphs show the percentages of C3 and GFAP (*n* = 7, vehicle group; *n* = 5, IGU group) (**b**), iNOS and Iba-1 (*n* = 5 per group) (**d**), S100A10 and GFAP (*n* = 5, vehicle group; *n* = 4, IGU group) (**f**), and Arg-1 and Iba-1 (*n* = 6, vehicle group; *n* = 5, IGU group) (**h**) double-positive areas in the spinal cords of IGU-treated and vehicle-treated mice. Significant differences were determined using an unpaired *t*-test. * *p* < 0.05; ns, not significant.

**Figure 4 biology-12-01217-f004:**
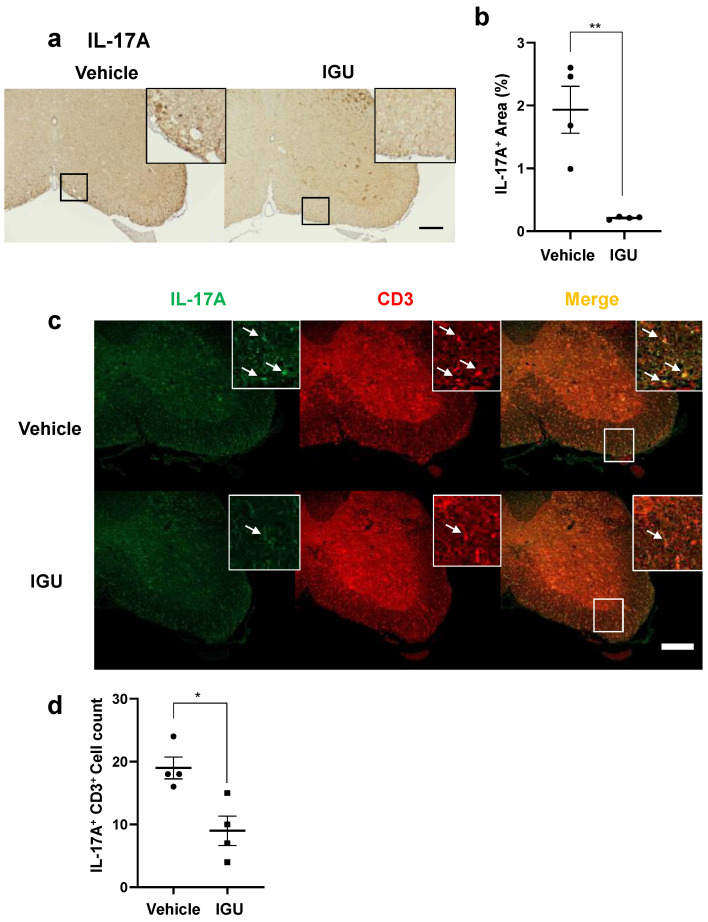
IGU suppressed T helper 17 (Th17) cell migration in the spinal cord during the chronic phase. (**a**) Interleukin (IL)-17A immunostaining in paraffin sections of the lumbar spinal cords of IGU-treated and vehicle-treated mice at 50 dpi. Scale bar: 100 µm. (**b**) The graph shows the percentages of IL-17A-positive areas in the spinal cords of IGU-treated and vehicle-treated mice (*n* = 4 per group). (**c**) Double immunofluorescence of the lumbar spinal cord for IL-17A and CD3 at 50 dpi. White arrows indicate IL-17A^+^ CD3^+^ Th17 cells. (**d**) The graph shows the number of IL-17A and CD3 double-positive cells in the spinal cords of IGU-treated and vehicle-treated mice (*n* = 4 per group). Scale bars: 100 µm. Significant differences were determined using an unpaired *t*-test. * *p* < 0.05; ** *p* < 0.01.

**Figure 5 biology-12-01217-f005:**
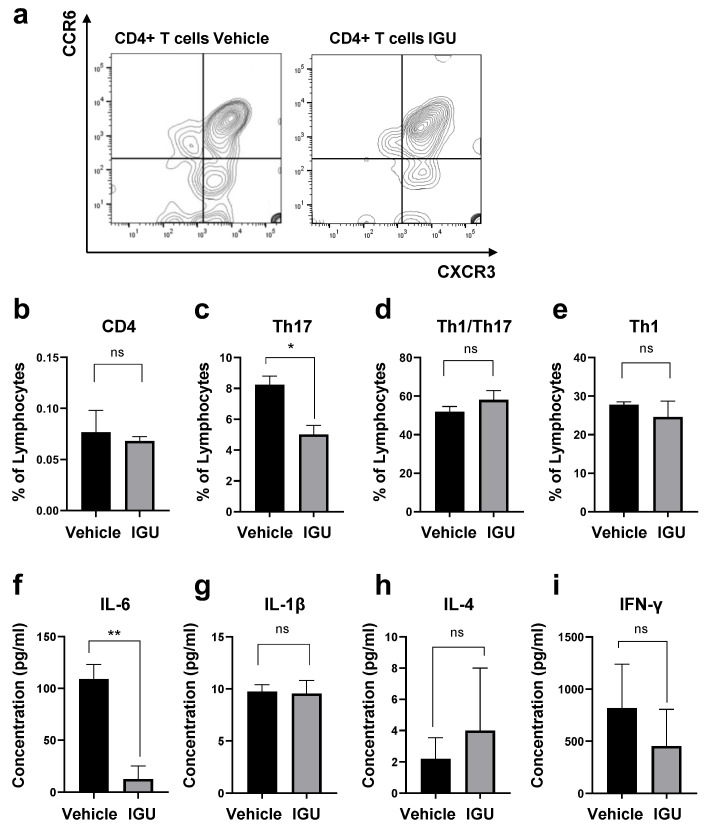
IGU inhibited cerebrospinal fluid (CSF) cytokine production and CD4^+^ T cell infiltration into the central nervous system (CNS) tissue (spinal cord and brain) of pEAE mice during the chronic phase. (**a**) Representative flow cytometry plots showing the gating strategy for T cells isolated from the CNS tissue of vehicle-treated (left side) and IGU-treated (right side) pEAE mice. (**b**) The percentages of CD4^+^ T cells among lymphocytes in the CNS tissue of pEAE mice at 50 dpi were measured by flow cytometry (*n* = 4 per group). (**c**–**e**) The percentages of CXCR3^−^ CCR6^+^ CD4^+^ Th17 (**c**), CXCR3^+^ CCR6^+^ CD4^+^ Th1/Th17 (**d**), and CXCR3^+^ CCR6^−^ CD4^+^ Th1 (**e**) cells among CD4^+^ T cells were measured by flow cytometry (*n* = 4 per group). (**f**–**i**) CSF cytokine levels in the chronic phase of pEAE. IL-6 (*n* = 4 per group) (**f**), IL-1β (*n* = 4, vehicle group; *n* = 5, IGU group) (**g**), IL-4 (*n* = 5 per group) (**h**), and interferon (IFN)-γ (*n* = 4 per group) (**i**) were analyzed using a multiplexed fluorescence immunoassay. Significant differences were determined using an unpaired *t*-test. * *p* < 0.05; ** *p* < 0.01; ns, not significant.

**Figure 6 biology-12-01217-f006:**
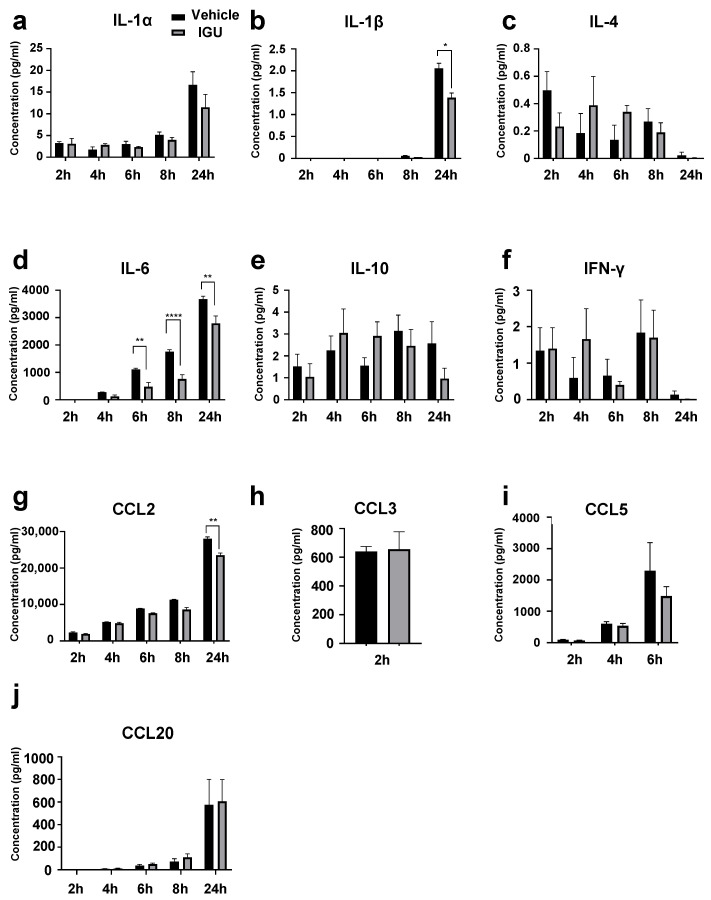
IGU inhibited cytokine/chemokine release from glial cells in vitro in an activated glial inflammation model. Cytokine levels in the culture supernatant were collected at each time point. (**a**) IL-1α levels (*n* = 4 per group). (**b**) IL-1β levels (*n* = 4 per group). (**c**) IL-4 levels (*n* = 4 per group). (**d**) IL-6 levels (*n* = 3 per group). (**e**) IL-10 levels (*n* = 4 per group). (**f**) IFN-γ levels (*n* = 4 per group). (**g**) CCL2 (MCP-1) levels (*n* = 5 per group). (**h**) CCL3 (MIP-1α) levels (*n* = 4 per group). Concentrations between 4 and 24 h were saturated (>4289 pg/mL). (**i**) CCL5 (RANTES) levels (*n* = 4 per group). Concentrations between 8 and 24 h were saturated (>62,700 pg/mL). (**j**) CCL20 (MIP-3α) levels (*n* = 4 per group). Significant differences were determined by two-way analysis of variance (ANOVA), followed by Sidak’s multiple comparisons test. * *p* < 0.05; ** *p* < 0.01; **** *p* < 0.0001; signs for not “significant (ns)” are omitted when there is no significant difference in the groups for each hour. Black bar: vehicle group. Gray bar: IGU group.

**Figure 7 biology-12-01217-f007:**
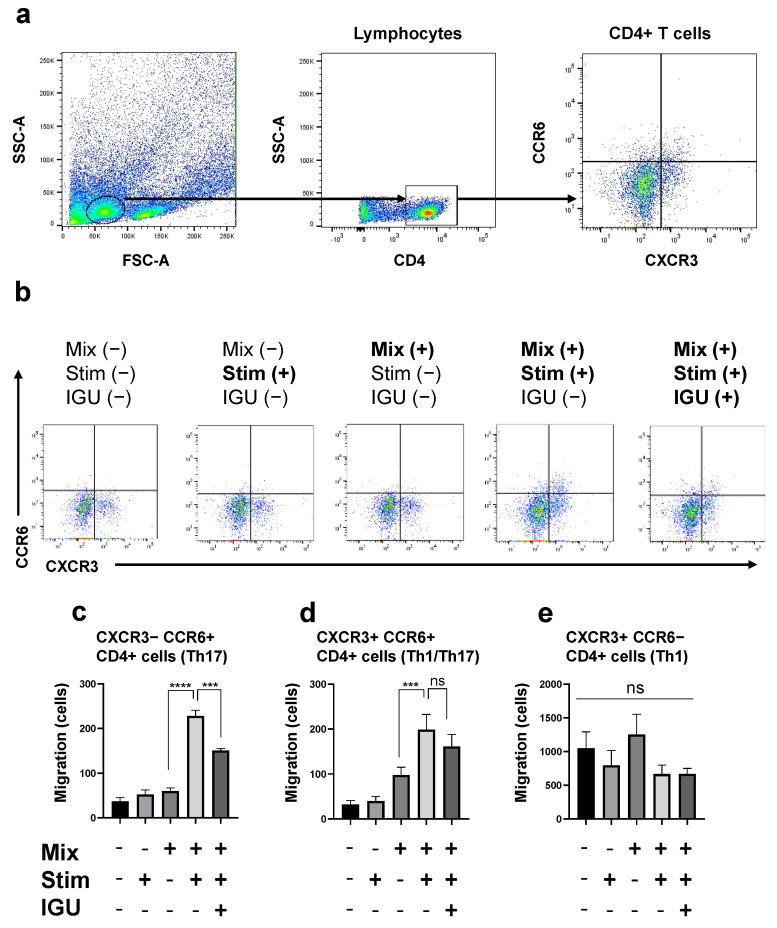
IGU decreased the number of migrating Th17 and Th1/Th17 cells in an in vitro migration assay. (**a**) Gating strategy for the evaluation of CD4^+^ T cells migrated from the upper chamber to the lower chamber. (**b**) The numbers of CXCR3^−^ CCR6^+^ CD4^+^ Th17 cells, CXCR3^+^ CCR6^+^ CD4^+^ Th1/Th17 cells, and CXCR3^+^ CCR6^−^ CD4^+^ Th1 cells that migrated from the upper chamber to the lower chamber with or without mixed glial cells (1.0 × 10^6^ cells/well), stimulation (IL-1α, C1q, tumor necrosis factor (TNF)-α, lipopolysaccharide (LPS)), and IGU were counted using flow cytometry. (**c**–**e**) Bar graphs showing the numbers of Th17 cells (**c**), Th1/Th17 cells (**d**), and Th1 cells (**e**) that migrated in each lower chamber (*n* = 4 per group). For clarity, in the graphs of Th17 cells and Th1/Th17 cells, signs of significance were used only for comparisons between Mix(+) Stim(−) IGU(−) and Mix(+) Stim(+) IGU(−) and between Mix(+) Stim(+) IGU(−) and Mix(+) Stim(+) IGU(−). Significant differences were determined by one-way ANOVA, followed by Tukey’s multiple comparisons test. *** *p* < 0.001; **** *p* < 0.0001; ns, not significant. Mix: mixed glial cell culture. Stim: stimulation. IGU: iguratimod.

## Data Availability

Not applicable.

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
