# Peer review of "Iguratimod Ameliorates the Severity of Secondary Progressive Multiple Sclerosis in Model Mice by Directly Inhibiting IL-6 Production and Th17 Cell Migration via Mitigation of Glial Inflammation"

_biology, 2023, doi:10.3390/biology12091217_

Round 1

Reviewer 1 Report

Dear Authors

In the present study, the researchers tried to investigate the effects of "IGU ameliorates the severity of ...." 

I enjoyed the manuscript. It was well-design and the writing was fluent. But there are some questions.

Line 111: what was the role of tamoxifen to create knockout mice?

Line 111: you stated that tamoxifen was injected at 8-10 weeks but in line 293 you mentioned 10-12. Which is correct?

Fig 1d, line 315: repetitive sentence: "scale bar: 100 micrometer". please delete it.

Fig 6: in footnote, line 423: you mentioned two way ANOVA but in line 288, you just stated about one way ANOVA or unpair t-test. Please complete statistical section.

In discussion, you mentioned the positive effects of IGU.

Were there any opposite (counter) opinions or reference/s to emphasize the negative effect of IGU? 

For improvement and much discussion, please add the counter comments or references in discussion section.

Reviewer 2 Report

The authors test the effect of iguratimod on MOG-induced EAE on inducible conditional Cx47 knockout (Cx47 icKO) mice. They had previously shown that iguratimod is effective in ameliorating MOG-induced EAE in the wild type C57BL/6J mice through inhibition of T cell infiltration and macrophage activation, attributable in part to inhibition of the NF-kB pathway. In this paper, the authors showed that treatment with iguratimod also reduces the clinical course of EAE in the Cx47 icKO mice with corresponding reduction in the extent of demyelination, glial inflammation and CNS infiltrating T cells. IL-6 was reduced by treatment with iguratimod. Supporting evidence was provided for the anti-trafficking effect of iguratimod on CCR6+CXCR3- cells which was interpreted as anti-trafficking effect on Th17 cells. 

Overall, this was a well designed, well conducted study that represents an incremental advance in knowledge. Experiments were properly controlled. Well-established methods were appropriately used. 

The following comments need to be addressed.

METHODS:

  • Was EAE clinical scoring done by personnel blinded to treatment allocation? If so, state this in the methods. If not, state that as a limitation of the study.

  • Please state how many cord sections per animal were analyzed for the quantification of immunohistochemical images

  • The use of chemokine receptor expression alone to identify T helper lineage of infiltrating CD4+ T cells in EAE remains somewhat controversial (see Mony, JT et al. Front Cell Neurosci. 2014). Results should indicate the actual markers detected rather than T helper lineage inferred (see comments below). 

  • Flow cytometry: absolute counts should be reported in addition to relative frequencies. If absolute counts were not determined, that should be reported as a limitation of the study in the discussion.

RESULTS:

  • Section 3.4 & 3.6: needs to be revised to avoid the assumption that CXCR3 and CCR6 expression alone can reliably identify Th1 and Th17 lineage cells. OK to make the inference that CXCR3-CCR6+ are Th17 cells and CXCR3+CCR6- are Th1 cells but need to explicitly state that the Th lineages are inferred from chemokine receptor expression.

  • Figure 5c to 5e Y axis appears to be mislabeled as “% lymphocytes”. Figure caption indicates Y axes are % of CD4+ T cells. Please revise or clarify.

  • Figure 5c to 5e should be re-labeled with the actual chemokine receptor detected (CXCR3, CCR6) rather than the inferred T helper lineage.

  • Figure 7c to 7e should be re-labeled with the actual chemokine receptor detected (CXCR3, CCR6) rather than the inferred T helper lineage.

  • Section 3.7: not clear how T helper lineage was identified in Figure S2. Please state in the Methods or Figure S2 caption.

DISCUSSION

  • State that Th1 and Th17 lineage differentiation of CD4+ T cells were identified not by lineage defining cytokine or transcription factor but inferred based on chemokine receptor expression

Reviewer 3 Report

This manuscript utilizes a novel model of EAE to examine the effects of Iguratimod (IGU) a putative cox2 inhibitor of the progression of disease. Previous studies have shown that selective depletion of Cx47 from oligodendrocytes results in a pronounced demyelinating pathology when coupled to EAE stimulation. Here the authors demonstrate that treatment at the peak of disease with IGU reduces disease burden dramatically. Immunohistichemistry studies suggest a reduction in glial reactivity, and a decrease in T cell infiltration following IGU treatment. In vitro studies implicate selected cytokines in driving disease which are compromised following IGU treatment.  There are some intersting aspects ot the paper. Other studies haves suggested IGU may be effective in modulating disease in EAE and demonstration of its effect in another model is useful. The authors use mutliple approahes to examine potential mechanisms that might mediate the effects and the data is quite clear. The major problem with this paper is the immunohistochemistry. All the figures, 1,2,3 and to a lesser extent 4 are not convincing and fail to provide confidence in the validity of the data. For example, the images in Fig 2 do not have sufficient contrast to clearly see labeled cells.  More worrisome the distribution of known cell types does not look as expected (GFAP should be far more pronounced in white matter). Likewise, in Fig 3 S100 looks like background staining. All the images need to be presented in a more convincing manner prior to consideration for publication.

Round 2

Reviewer 3 Report

The authors have significantly improved the quality of the IHC data and the paper is considerably stronger as a consequence.

The work is interesting and the result quite convincing.

The use of English is satisfactory